# Utilizing the Off-Target Effects of T1R3 Antagonist Lactisole to Enhance Nitric Oxide Production in Basal Airway Epithelial Cells

**DOI:** 10.3390/nu15030517

**Published:** 2023-01-19

**Authors:** Derek B. McMahon, Jennifer F. Jolivert, Li Eon Kuek, Nithin D. Adappa, James N. Palmer, Robert J. Lee

**Affiliations:** 1Department of Otorhinolaryngology, University of Pennsylvania Perelman School of Medicine, Philadelphia, PA 19104, USA; 2Department of Physiology, University of Pennsylvania Perelman School of Medicine, Philadelphia, PA 19104, USA

**Keywords:** T1R1, T1R3, umami, lactisole, apoptosis, nitric oxide, airway

## Abstract

Human airway sweet (T1R2 + T1R3), umami (T1R1 + T1R3), and bitter taste receptors (T2Rs) are critical components of the innate immune system, acting as sensors to monitor pathogenic growth. T2Rs detect bacterial products or bitter compounds to drive nitric oxide (NO) production in both healthy and diseased epithelial cell models. The NO enhances ciliary beating and also directly kills pathogens. Both sweet and umami receptors have been characterized to repress bitter taste receptor signaling in healthy and disease models. We hypothesized that the sweet/umami T1R3 antagonist lactisole may be used to alleviate bitter taste receptor repression in airway basal epithelial cells and enhance NO production. Here, we show that lactisole activates cAMP generation, though this occurs through a pathway independent of T1R3. This cAMP most likely signals through EPAC to increase ER Ca^2+^ efflux. Stimulation with denatonium benzoate, a bitter taste receptor agonist which activates largely nuclear and mitochondrial Ca^2+^ responses, resulted in a dramatically increased cytosolic Ca^2+^ response in cells treated with lactisole. This cytosolic Ca^2+^ signaling activated NO production in the presence of lactisole. Thus, lactisole may be useful coupled with bitter compounds as a therapeutic nasal rinse or spray to enhance beneficial antibacterial NO production in patients suffering from chronic inflammatory diseases such as chronic rhinosinusitis.

## 1. Introduction

In humans, the taste perception of either sweet or savory/umami chemicals occurs through type 1 taste receptors (T1Rs). The perception of sweet taste signals the presence of beneficial energy-rich sugars, and this pathway is thought to be signaled by the interaction of sweet tasting molecules with a G protein-coupled receptor (GPCR) heterodimer of T1R2 and T1R3 [1,2] localized to taste buds on the tongue. Alternatively, the sensation of umami signals the presence of beneficial amino acids, and umami signaling is initiated via a proposed heterodimer of T1R1 and T1R3 [3]. Both pathways share the common component of T1R3 and signal through intracellular Ca^2+^ elevations. T1R3 has also been proposed to signal independently of T1R1 or T1R2, either as a homodimer or with another GPCR partner [4,5,6,7,8,9]. T1Rs serve as nutrient detectors on the tongue and also throughout the body, including in glucose sensing in pancreatic β-cells [5,6,7,8,9] and adipocytes [4]. Pharmacologically regulating nutrient detection of T1Rs have been proposed as a potential therapeutic strategy for diabetes [5,6,7,8,9]. While multiple artificial sweeteners have been developed that can activate T1R2 and/or T1R3 [1,2], experimental tools to inhibit T1R3 nutrient detection are more limited. 

Experiments where T1R nutrient sensing is inhibited often employ lactisole, a molecule isolated from coffee beans [10]. Lactisole has been shown to inhibit T1R3 through interactions with a binding pocket within T1R3′s transmembrane domain [11]. Once bound, lactisole inhibits the downstream Ca^2+^ pathways activated by the detection of sweet [12] or umami molecules [13]. Consequently, in human taste signaling pathways, lactisole inhibits the detection of both sweet and savory molecules. In addition to biochemical and psychophysical research studies, lactisole is also used in the food industry in high-sugar-content foods such as fruit jams or jellies to reduce sweetness and allow other flavors to be perceived [10,11,12,13]. However, while the molecular interactions of lactisole with T1R3 have been examined in detail [11], to our knowledge, off-target effects of lactisole have not been investigated.

Apart from their function in taste signaling pathways, T1Rs are also expressed in the human upper airway, where they function with bitter type 2 taste receptors (T2Rs) to detect pathogenic growth. T2Rs are localized to the cilia of multi-ciliated epithelial cells where they detect “bitter” bacterial byproducts, such as quorum sensing molecules. Once activated, T2Rs signal through intracellular Ca^2+^ to stimulate nitric oxide (NO) production [14,15,16] and an increase in ciliary beat frequency [15,16,17], increasing the mucociliary clearance of pathogens out of the airway. T2Rs are also found on airway solitary chemosensory cells (SCCs) [18,19] along with the sweet taste receptor components T1R2 and T1R3 [19]. There, T1R2 and T1R3 detect apical levels of glucose and function to repress T2R signaling pathways [19]. However, once pathogen growth begins to deplete apical glucose levels, the repression that T1R2 and T1R3 elicit on T2R signaling is relieved, allowing T2Rs to signal via intracellular Ca^2+^ within the SCC to trigger a release of Ca^2+^, acetylcholine, and other signaling molecules through gap junctions into surrounding multi-ciliated epithelial cells [19]. These signaling molecules then cause antimicrobial peptide release into the apical lumen, killing pathogens [19]. While mucus provides a physical barrier to ensnare pathogens, the downstream signaling pathways of T1Rs and T2Rs initiate an innate immune response to both kill and accelerate the removal of pathogens from the airway.

Viruses such as influenza [20] or SARS-CoV-2 [21] and chronic inflammatory diseases such as chronic rhinosinusitis [22] or severe asthma [23] cause a remodeling of the airway epithelium leading to ciliary dysfunction or a complete loss of multi-ciliated epithelial cells, instead being replaced by basal epithelial cells lacking motile cilia. While these cells lack the innate defenses provided by motile cilia, we recently found that they maintain expression of T2Rs and retain the ability to initiate NO production [24]. We have also previously shown that T1Rs are expressed in cultured basal airway epithelial cells and function to detect amino acids through cAMP signaling pathways and, when stimulated by amino acids, function to repress T2R Ca^2+^ signaling pathways through reducing total ER Ca^2+^ levels [25]. Thus, we hypothesized that in airway epithelial cells, by inhibiting T1R3 with lactisole, we would increase T2R-stimulated NO production.

Here, we show that lactisole increases intracellular cAMP levels independent of T1R1 or T1R3 expression. Additionally, lactisole both increases ER Ca^2+^ content and denatonium-induced cytosolic Ca^2+^ elevations, ultimately signaling an increase in NO production. We propose here that these off-target effects of lactisole may be therapeutically useful for boosting airway NO production. However, our data also suggest that studies using lactisole to experimentally or therapeutically inhibit T1R sugar or amino acid detection must take into account the potential effects of T1R-independent cAMP signaling and/or changes in ER Ca^2+^ signaling that may also occur.

## 2. Materials and Methods

A list of all reagents used in the current study are shown in Appendix A.

### 2.1. Live Cell Imaging

To assess intracellular Ca^2+^, cells were incubated with 5 µM of Fluo-8 AM for 1 h in Hank’s Balanced Salt buffered with 20 mM HEPES free acid. Fura-2 AM (5 µM) was utilized for assessment of baseline intracellular Ca^2+^ levels. Images of cells loaded with Fura-2 were captured using Fura-2 filters (79002-ET Chroma, Rockingham, VT, USA). To assess NO production, cells were loaded with 10 µM DAF-FM diacetate for 45 min. For intracellular or nuclear cAMP, nuclear, mitochondrial, or non-nuclear Ca^2+^ measurements, cells were transfected with the appropriate biosensors 48 h prior to experiments. Images taken with either FITC or TRITC utilized 49002-ET or 49004-ET Chroma respectively. All images were taken on an Olympus IX-83 microscope (20 × 0.75 NA objective) with excitation and emission filter wheels from Sutter Instruments (Novato, CA, USA), an Orca Flash 4.0 sCMOS camera (Hamamatsu, Tokyo) and used MetaFluor (Molecular Devices, Sunnyvale CA, USA) software. Light sources included a xenon lamp for Fura-2 and a X-Cite 120 boost LED source (Excelitas, Waltham, MA, USA) for all other imaging.

### 2.2. Culture of Primary Human Cells

Residual human surgical material was used a source for isolating primary sinonasal cells. Samples were obtained conforming to the University of Pennsylvania guidelines for use along with the Declaration of Helsinki and the U.S. Department of Health and Human Services code of federal regulation (Title 45 CFR 46.116). Institutional review board approval (#800614) was obtained, and informed consent was collected from each patient. Patients were adults, >18 years old, and underwent surgery to address a sinonasal disease or for other causes.

Unless otherwise noted, the primary nasal epithelial cells were isolated from the residual surgical material via enzymatic digestion as previous described [14]. In short, the tissue was incubated for 1 h at 37 °C in digestion medium containing 1.4 mg/mL protease and 0.1 mg/mL DNase. Proteases were inactivated by the addition of medium supplemented with FBS (10%). To remove non-epithelial cells, the resulting cells were incubated in a flask for 2 h at 37 °C with 5% CO_2_ in PneumaCult-Ex Plus medium (Cat. 05040, Stemcell Technologies, Vancouver, Canada) supplemented with penicillin (100 U/mL) and streptomycin (100 ug/mL). After 2 h had passed, the media were then transferred to a cell culture dish and the resulting cultures were expanded utilizing PneumaCult-Ex Plus medium. To obtain fully differentiated cultures, cells were seeded in air–liquid interface (ALI) cultures and exposed to air for at least 21 days prior to use in experiments.

Primary bronchial epithelial cells were purchased through Lonza (CC-2540S) and propagated and differentiated using the same methods as described for primary nasal epithelial cells.

### 2.3. Knockdown of T1Rs

Subconfluent cultures of Beas-2B cells (propagated in F-12K Media with 10% FBS, 1% penicillin/streptomycin) were transfected via lipofectamine 3000 for 48 h with either 10 nM of TAS1R1.6 or TAS1R3.2 or control non-targeting RNAi duplex (Integrated DNA Technologies, Coralville, IA, USA). Validation of knockdown was achieved via qPCR analysis.

### 2.4. Quantitative PCR (qPCR)

RNA was collected by resuspending cultures in TRIzol (ThermoFisher Scientific, Waltham, MA USA). Suspensions were used immediately or stored at 70 °C for later use. The Direct-zol RNA kit (Zymo Research, Irvine, CA, USA) was used to purify the preserved RNA. cDNA libraries were created using RT-PCR using the High-Capacity cDNA Reverse Transcription Kit (ThermoFisher Scientific). Resulting cDNA libraries were then subjected to qPCR using Taqman qPCR probes and analyzed via the QuantStudio 5 Real-Time PCR System (ThermoFisher Scientific). Microscoft Excel and GraphPad PRISM v8 were used to both analyze and plot the resulting data.

### 2.5. Genotyping T2R38 PAV/AVI

Unless otherwise noted, genotyping methods followed as previously described [26]. Samples were vortexed briefly then centrifuged in a tabletop centrifuge and max speed for 5 min. Aqueous phase was transferred to a new tube and 3M Sodium Acetate pH 5.2 was added at 1/10th the volume followed by two volumes of isopropanol. Samples were centrifuged in a tabletop centrifuge at max speed for 40 min at 4 °C. Supernatant was decanted and resulting pellet was washed twice with 1 mL 70% ethanol then air dried at room temperature before resuspension in 100 μL of Tris-EDTA buffer. Resulting DNA was subjected to PCR via TaqBlue in reactions utilizing 200 μM dNTPs, 250 μM Forward (TTG GGA TAA TGG CAG CTT GTC CCT C), 250 μM Reverse (GCA CAG TGT CCG GGA ATC), 0.05 U/μL TaqBlue, and 1–20 ng/μL DNA template. Resulting PCR products were digested using 10× CutSmart Buffer and 0.5 μL FNu4HI at 37 °C for 1 h then separated on a 2.2% agarose gel containing ethidium bromide. Cleaved DNA bands were imaged via the ChemiDoc MP Imaging System (BioRad Laboratories, Hercules, CA, USA).

### 2.6. Data Analysis and Statistics

For graphs with just two comparisons, *t*-tests were utilized. Graphs or plots with >2 comparisons utilized the ANOVA test, whereas the Tukey–Kramer post-test was used for comparing multiple samples to each other, the Bonferroni post-test was used for selective pair comparisons, Sidak’s post-test for paired comparisons, or Dunnett’s post-test for multiple comparisons relative to one control. In all comparisons, (*) *p* < 0.05, (**) *p* < 0.01, (***) *p* < 0.001, (****) *p* < 0.0001, “n.s.” (no statistical significance). All data presented are the mean ± SEM of at least 3 experiments.

## 3. Results

### 3.1. Lactisole Increases Intracellular cAMP

Both sweet and umami taste signaling pathways are thought to share a common component, the GPCR T1R3. In humans, lactisole inhibits the perceived taste of both sweet [12] and umami [13] compounds at concentrations of >1 mM. Lactisole binds to the transmembrane domains of T1R3 where it interrupts T1R3-mediated Ca^2+^ signaling [11,27,28]. Previously, we have shown that the sweet taste receptor represses T2R signaling pathways in primary differentiated airway cultures [19] and the umami receptor regulates T2R signaling pathways in basal airway epithelial cells [25]. Therefore, we hypothesized that the addition of lactisole may alter T2R-signaled NO production through T1R3 inhibition.

We previously showed that both T1R1 and T1R3 respond to amino acids and induce cAMP elevations in airway basal epithelial cells [25]. While attempting to inhibit T1R activity with lactisole, we found that lactisole activated cAMP generation in a dose-dependent manner from 5 to 40 mM in Beas-2Bs expressing cAMP biosensor Flamindo2 (Figure 1a). In Beas-2Bs loaded with Ca^2+^ detection dye Fluo-8 AM, lactisole alone had no effect on intracellular Ca^2+^ (Figure 1b). Likewise, lactisole also did not alter baseline intracellular Ca^2+^ levels as measured via ratiometric Ca^2+^ detection dye Fura-2 (Figure 1c). Lactisole treatment activated both nuclear and non-nuclear cAMP pathways (Figure 1d) and persisted for ≥1 h after treatment (Figure 1e). Interestingly, in pancreatic β cells, while lactisole inhibited sucralose-induced Ca^2+^ release, it did not diminish cAMP elevations signaled by sweet compounds [29]. Therefore, we hypothesized that this alteration of cAMP signaling might be a poorly understood function of lactisole, either via T1R3 or in a T1R3-independent manner.

To determine if lactisole was signaling cAMP elevations through T1R3, we utilized a T1R3 knockdown model in Beas-2Bs. Using RNAi duplexes to reduce T1R3 and T1R1 expression by ~70% [25], we observed similar levels of lactisole-mediated intracellular cAMP elevations in knockdown cells relative to cells expressing non-targeting RNAi (Figure 1f,g). Thus, lactisole activated cAMP independent of T1R3 and T1R1. Together, these data suggest that cAMP elevation may be an off-target effect of lactisole. However, given the elevation of cAMP with lactisole, we further investigated the downstream effect of signaling pathways activated by lactisole.

### 3.2. Lactisole Increases ER Ca^2+^ Content, ER Ca^2+^ Efflux, and GPCR-Modulated Ca^2+^ Signaling

Even though lactisole did not stimulate Ca^2+^ release, it could still modify Ca^2+^ signaling in other ways, such as through a modification of ER Ca^2+^ store content. Because most GPCR Ca^2+^ release originates with activation of phospholipase C and production of IP_3_, changes in ER Ca^2+^ stores can alter GPCR Ca^2+^ release dynamics by modifying the driving force for IP_3_ receptor-mediated Ca^2+^ efflux from the ER. To quantify ER Ca^2+^ content, we loaded Beas-2Bs with Fluo-8 AM and inhibited the SERCA pumps using thapsigargin (10 µg/mL) in 0-Ca^2+^ HBSS with 2 mM EGTA. By inhibiting ER Ca^2+^ uptake, the total ER Ca^2+^ stores slowly leaked and caused Fluo-8 to fluoresce before continuing to disperse out of the cell and chelating to extracellular EGTA, preventing re-uptake. Beas-2B cells treated with 20 mM lactisole for 1 h revealed a 275% increase in ER Ca^2+^ release as observed through comparisons of peak Ca^2+^ elevations and a 200% increase in ER Ca^2+^ stores as calculated by the area under the curve (Figure 2a). While calculating the area under the curve for each trace allows for an observation of total ER Ca^2+^ content, calculating the initial linear slope of Ca^2+^ release allows for an estimation of the rate of Ca^2+^ efflux (leak) from the ER. As seen in Figure 2b, pretreatment of Beas-2Bs with lactisole (20 mM, 1 h) caused a 300% increase in the magnitude of the initial linear slope in Fluo-8 fluorescence. Together, these results demonstrate that lactisole functioned to both increase ER Ca^2+^ content and the rate of Ca^2+^ efflux from the ER. With such a dramatic increase in ER Ca^2+^ stores, we wanted to investigate if the ER itself increased in size.

To determine if the ER physically increased in size in response to 20 mM lactisole treatment, Beas-2B cells were loaded with 2 µM of ER-Tracker Green, a live-cell stain that binds to the sulfonylurea receptors of ATP-sensitive K^+^ channels prominent on the ER, to visualize the ER’s morphology. Here, we show that in live cells loaded with ER-Tracker, there is a 25% reduction in fluorescence signal in cells pretreated with 20 mM lactisole for 1 h relative to untreated cells (Figure 2c). In just one hour, it is unlikely that the cells would be able to produce more ATP-sensitive K^+^ channels to stain. It is most likely that in that time frame the expansion of the ER caused more distance between these channels, and therefore a decrease in overall fluorescence intensity. Together with our measurements of ER Ca^2+^ content, these data suggest that lactisole treatment increased both the Ca^2+^ content and size of the ER.

Next, we utilized the ER-localized Ca^2+^ biosensor D1ER [30] to examine if the increase in ER Ca^2+^ stores led to an increase in ER Ca^2+^ efflux via GPCR-signaled pathways. We observed that in Beas-2B cells expressing D1ER, denatonium treatment caused a greater decrease in ER Ca^2+^ levels in cells treated with lactisole compared to those which were untreated (Figure 2d). This change in ER Ca^2+^ release was approximately a 250% increase relative to the untreated control. To definitively answer whether extracellular Ca^2+^ had any influence on this lactisole-mediated increase in denatonium-induced Ca^2+^ release, Beas-2Bs were preincubated with 20 mM lactisole for 1 h then stimulated with 15 mM denatonium in the presence or absence of extracellular calcium. We observed that there was no difference in denatonium-induced peak Ca^2+^ release whether extracellular Ca^2+^ was present or absent (Figure 2e). Cumulatively, these data suggest that the increase in denatonium-induced Ca^2+^ release from lactisole pretreatment originated solely from the ER.

We also tested if a similar effect on Ca^2+^ signaling was observable using other bitter compounds, each which activate a variety of T2Rs (Appendix A). Compared to other bitter compounds, 20 mM lactisole pretreatment (1 h) had the highest impact on denatonium-induced Ca^2+^ signaling with a 250% uplift, while 500 µM flufenamic acid, 1 mM quinine, or 3 mM thujone increased Ca^2+^ elevations by 30% (Appendix A). Interestingly, 5 mM diphenhydramine uniquely displayed a 175% and 200% increase with 10 or 20 mM lactisole pretreatment respectively (Appendix A). We have previously reported that denatonium Ca^2+^ signaling pathways were inhibited by a 1 h pretreatment with 1 µM of Gq inhibitor YM-254890 [24]. Here, we show that denatonium is the only bitter compound of the ones tested here that was sensitive to YM-254890 (Appendix A). These data suggest that the dramatic increase in denatonium-induced Ca^2+^ response may be likely due to its associated T2R(s) signaling via a Gq. Moreover, we only observed a denatonium-induced Ca^2+^ release in airway cells line RPMI-2650 (Appendix A) and primary basal NHBE cells **(**Appendix A) with lactisole pretreatment. Interestingly, differentiated NHBE cells did not signal via Ca^2+^ in response to denatonium regardless of lactisole pretreatment (Appendix A). Differentiated NHBE cells do signal through Ca^2+^ in response to thujone (3 mM), however this signaling was unaffected by lactisole pretreatment (Appendix A). Together these observations in NHBE cells suggest that differentiated cultures may lack the protein(s) found in basal cells that lactisole is signaling. Together, these data demonstrate that this effect is unique to basal airway epithelial cells.

We also tested the ability of Beas-2Bs to detect phenylthiocarbamide (PTC), a bitter compound associated with T2R38 [31]. In humans, T2R38 and its variations have been well characterized. The PAV (P49, A262, and V296) variation is fully functional and capable of strongly tasting PTC [32]. However, when these amino acids are mutated to AVI respectively, this causes an inability to taste PTC [32]. In fully differentiated airway cultures, PTC activates intracellular Ca^2+^ pathways in cultures expressing the PAV-version of T2R38 [14]. We genotyped Beas-2Bs as having heterozygous PAV/AVI *TAS2R38* expression (Appendix A). While Beas-2Bs displayed a low level of intracellular Ca^2+^ elevation when treated with 5 mM PTC, lactisole pretreatment (1 h, 20 mM) increased PTC-induced Ca^2+^ release by 300% (Appendix A). Utilizing Beas-2Bs expressing either nuclear or non-nuclear variations of R-GECO, we observed that these Ca^2+^ elevations were in nuclear and non-nuclear compartments (Appendix A). Thus, lactisole was able to bolster this Ca^2+^ signaling pathway in Beas-2Bs, which is partially hindered by heterozygous PAV/AVI TAS2R38 expression.

We also found that lactisole effect on Ca^2+^ signaling pathways was not unique to T2R agonists. Histamine-induced Ca^2+^ release was also increased with lactisole pretreatment (Figure 2f), further supporting our hypothesis that lactisole itself does not directly modify T2R function but acts to amplify Ca^2+^ signaling pathways through increasing ER Ca^2+^ content. We then wanted to explore if lactisole mediated these changes in ER Ca^2+^ storage through cAMP elevations.

### 3.3. EPAC Increases ER Ca^2+^ Efflux

Elevations in cAMP can activate downstream signaling proteins such as Protein Kinase A (PKA) or Exchange Protein Directly Activated by cAMP (EPAC). Both have been shown to affect Ca^2+^ signaling. Using Beas-2Bs expressing ratiometric PKA biosensor AKAR4 [33,34], we did not observe changes in intracellular cAMP with 20 mM lactisole (Figure 3a). Additionally, 20 mM of lactisole did not alter nuclear PKA activity in Beas-2B cells expressing nuclear localized AKAR4-nls [34] (Figure 3b). Even after an hour preincubation with 20 mM lactisole, there were no observed changes in baseline intracellular PKA activity (Figure 3c). Therefore, if changes are occurring in intracellular PKA, they are beyond the detection limits of this biosensor. We have previously shown that treatment with bitter compounds reduced resting cAMP levels [24]. Beas-2Bs pretreated with 10 µM of PKA inhibitor H89 for one hour demonstrated reduced denatonium-induced Ca^2+^ elevations (Figure 3d). Interestingly, treatment with H89 also completely blocked lactisole’s ability to increase intracellular Ca^2+^ elevations (Figure 3d). These data reveal that baseline cAMP levels and PKA activity plays a very potent role in Ca^2+^ signaling pathways. However, given our findings with PKA sensor AKAR4, PKA pathways are most likely not impacted by lactisole treatment.

Though PKA activity was not likely elevated by lactisole, we wanted to test if EPAC activity was affected. Using the ratiometric cAMP Epac-S-H74 [35] biosensor we found a mild but persistent increase in cAMP (Figure 4a). While this biosensor does not directly measure EPAC activity, much like the flamindo2 biosensor [36], it utilizes the cAMP binding site of EPAC to measure cAMP levels. An association between EPAC activity with ER Ca^2+^ efflux and influx pathways via modulation of SERCA and RYR Receptors to create a Ca^2+^-induced Ca^2+^ release pathway [37,38,39,40]. Due to the lack of PKA activation by lactisole, we further explored the hypothesis that lactisole-signaled cAMP may be signaling through EPAC to alter ER Ca^2+^ content.

To better understand ER Ca^2+^ dynamics in airway epithelial cells, we first utilized qPCR to determine the expression levels of inositol 1,4,5-triphosphate (IP3) receptors, ryanodine receptors, and sarcoendoplasmic reticulum calcium ATPase (SERCA) pump expression. Here, we show that in Beas-2Bs, primary basal, and primary differentiated airway epithelial cells express all forms of IP3 receptors (*ITPR1-3*) and SERCA pumps (*ATP2A1-3*) however they lack expression of ryanodine receptor 2 (*RYR2*) (Appendix A). Additionally, ryanodine receptor agonist caffeine (10 mM) did not elevate intracellular Ca^2+^ levels in Beas-2B cells (Appendix A), supporting our observations of low levels of ryanodine receptor transcript. Together, these data suggest that, apart from lacking *RYR2* expression, Beas-2Bs contain many of the ER-related proteins responsible for Ca^2+^ influx and efflux that are affected by EPAC activity.

Using thapsigargin to measure ER Ca^2+^ content as described above in Section 3.2, we found that with a 1 h treatment of 1 μM of EPAC agonist 8-pCPT-2′-O-Me-cAMP-AM (abbreviated 2′-O-Me-cAMP) there was a significant increase in Ca^2+^ efflux from the ER as observed through increases in the peak Ca^2+^ elevations and the relative magnitude of the intial linear slope in Fluo8 fluorescence relative to untreated cells, however there was no impact on total ER Ca^2+^ content (Figure 4b). Alternatively, 1 μM of EPAC antagonist ESI-09 decreased ER Ca^2+^ efflux and stores (Figure 4b). These data demonstrate that EPAC activity is a regulator of ER Ca^2+^ content in airway epithelial cells. However, activation of EPAC above baseline levels only increase ER Ca^2+^ efflux, not total ER Ca^2+^ storage.

Consistent with an increase in ER Ca^2+^ content, pretreatment of Beas-2Bs with 1 μM of EPAC agonist, 2′-O-Me-cAMP, revealed a 1.8-fold increase in denatonium-induced Ca^2+^ release and shortened the time of peak Ca^2+^ release by ~30% (Figure 4c). The increase in peak Ca^2+^ release per culture could be due to cells elevating Ca^2+^ in unison relative to untreated cultures that respond more sporadically [25]. To gain a better understanding of the total Ca^2+^ release for each individual cell, the peak Ca^2+^ release per cell was also analyzed independent of time. Treatment with 2′-O-Me-cAMP (1 µM for 1 h) increased the peak Ca^2+^ release per cell by 65% (Figure 4c). These results were surprising as 1 μM of 2′-O-Me-cAMP did not increase ER Ca^2+^ stores. To determine if this increase in Ca^2+^ elevation was originating from extracellular Ca^2+^, we repeated the experiment seen in (Figure 4c) but in 0-Ca^2+^ HBSS with 2 mM EGTA. As seen in Figure 4d, in cultures lacking extracellular Ca^2+^, 1 μM 2′-O-Me-cAMP does not significantly alter peak Ca^2+^ release per cell or reduce the time to peak Ca^2+^ release. These results suggest that stimulation of EPAC above baseline may also activate pathways used in extracellular Ca^2+^ influx. Alternatively, EPAC antagonist ESI-09 (1 µM for 1 h) blocked lactisole’s ability to increase denatonium-stimulated Ca^2+^ elevations (Figure 4e). Overall, these data reveal that EPAC activity plays an important role in regulating ER Ca^2+^ content and efflux. EPAC may also alter subsequent Ca^2+^ signaling pathways in airway epithelial cells through the uptake of extracellular Ca^2+^. Furthermore, we hypothesize that lactisole may be regulating ER Ca^2+^ levels through a pathway that is not solely reliant upon EPAC activity.

### 3.4. Lactisole Increases Denatonium-Induced Cytosolic Ca^2+^ to Activate NO Production

We have shown above that EPAC activation can increase intracellular Ca^2+^ release from T2R agonist denatonium by increasing not only the Ca^2+^ released per cell but by also reducing the time that Beas-2Bs reached peak Ca^2+^ release. To determine if lactisole had a similar effect, we pretreated Beas-2Bs with 10 mM of lactisole for 1 h then observed denatonium stimulated Ca^2+^ release via Fluo-8. Pretreatment of Beas-2Bs with lactisole amplified the levels of Ca^2+^ release by denatonium (Figure 5a). We also noted that the addition of lactisole doubled the number of cells responding to the denatonium treatment, doubled the Ca^2+^ release per responsive cell, and reduced the time it took to reach peak Ca^2+^ release by 1.5 min (Figure 5a). Both lactisole and 2′O-Me-cAMP appeared to increase the rate of ER Ca^2+^ efflux. Therefore, EPAC activity may play a role, at least in part, in lactisole’s ability to increase the rate of Ca^2+^ efflux from the ER, however there are likely other pathways involved.

While lactisole functioned to increase Ca^2+^ release via GPCR signaling pathways, we tested if sweet compounds would also have an effect. In differentiated airway epithelial cells, apical glucose concentrations were shown to repress the bitterant-induced T2R Ca^2+^ signaling pathway in solitary chemosensory cells [19]. Here, we show that in basal epithelial cells, denatonium Ca^2+^ signaling remained unaltered in HBSS lacking glucose or in high (25 mM) glucose conditions (Figure 5b). Sucralose is an artificial sweetener about 320–1000 times sweeter than sucrose and had no effect on denatonium-induced Ca^2+^ release regardless of 10 mM lactisole pretreatment (Figure 5c). These data suggest that while T1R2 transcript is detectable in airway epithelial cells [25], if the protein is expressed, its function is independent of what we observe with lactisole treatment here. Having determined that lactisole was not operating through a T1R in regard to its impact on Ca^2+^ signaling, we then further explored the role that lactisole has on denatonium’s downstream Ca^2+^ signaling pathways.

Previously, we observed denatonium signaled nuclear Ca^2+^ elevations in a dose-dependent manner in Beas-2B cells ranging from 5–25 mM, with maximal Ca^2+^ elevations beginning at 15 mM [24]. Here, utilizing Fluo-8, we show that intracellular Ca^2+^ release follows a similar dose-dependency as seen with nuclear Ca^2+^; 5 mM denatonium minimally elevated intracellular Ca^2+^ while concentrations above 15 mM reached maximal Ca^2+^ release (Figure 5d). The addition of 10 mM lactisole did not shift this dose-dependency curve either right or left, but instead elevated the Ca^2+^ release at all levels of denatonium capable of signaling detectable levels of Ca^2+^. Concentrations of 10, 15, or 20 mM denatonium had significantly higher Ca^2+^ elevations with the addition of 10 mM lactisole (Figure 4d). Using 15 mM denatonium to represent maximal Ca^2+^ elevation, lactisole was then titrated from 0–40 mM and revealed to reach maximal effect on denatonium induced Ca^2+^ release at a concentration of 20 mM (Figure 4e). Given these findings, we hypothesized that lactisole does not directly alter T2Rs (e.g., through post translation modifications) to increase Ca^2+^ efflux.

As visualized in Figure 6a, denatonium treatment (15 mM) activates both nuclear and mitochondrial Ca^2+^ elevations in basal airway epithelial cells, which we’ve previously shown to initiate mitochondrial membrane depolarization and caspase activity causing apoptosis [24]. To determine if the increased Ca^2+^ release observed with lactisole pretreatment increased the rate of the apoptosis, Beas-2Bs were treated with an intermediate dosage of 10 mM denatonium with or without a 1 h pretreatment with 20 mM lactisole. Using either CellEvent to quantify caspase activity (Figure 6b) or propidium iodide to measure cell death (Figure 6c), we did not observe any changes in apoptosis or cell death with lactisole pretreatment.

Consistent with these findings, using mitochondrial Ca^2+^ biosensor 4mtD3cpv [41], we also did not observe an increase baseline mitochondrial Ca^2+^ levels (Figure 6d) with lactisole pretreatment (20 mM, 1 h) or changes in denatonium-induced Ca^2+^ release due to lactisole pretreatment in Beas-2B cells (Figure 6e). Additionally, using nuclear Ca^2+^ biosensor nls-R-GECO [42], we did not observe any changes in nuclear Ca^2+^ signaling, however we did observe a 3-fold increase in non-nuclear Ca^2+^ release via a non-nuclear localized nes-R-GECO biosensor (Figure 6f). We have previously shown that denatonium mildly activates cytosolic Ca^2+^ [24]. Here, we show that lactisole amplified denatonium’s cytosolic Ca^2+^ signaling while leaving the nuclear and mitochondrial Ca^2+^ signaling pathways unaffected.

Previously, we observed that bitter compounds increase NO production in basal airway epithelial cells [24]. NO production is typically signaled via intracellular Ca^2+^ signaling pathways [43]. To determine if lactisole’s ability to increase cytosolic Ca^2+^ release altered NO production, we treated primary basal airway epithelial cells with NO detection dye DAF-FM with or without the presence of lactisole (20 mM, 45 min pretreatment). Primary basal epithelial cells produced NO when in the presence of lactisole (Figure 6g). Together, these data reveal that in airway epithelial cells, lactisole specifically increases denatonium-induced cytosolic Ca^2+^ signaling pathways necessary for NO production.

The above experiments have utilized acute treatments of lactisole. We also decided to evaluate the long-term effects of lactisole treatment. Over a 24 h treatment in Beas-2Bs, lactisole maintained an increased ER Ca^2+^ content (Appendix A). However, we also observed a slower rate of metabolism as measured by the XTT assay (Appendix A) and a slower growth rate measured by crystal violet staining (Appendix A). Additional studies will be necessary to explore the mechanism behind this reduction of growth; however, these results also suggest that lactisole’s elevation of ER Ca^2+^ is not transient.

Overall, we have shown that lactisole elevates cAMP levels independent of umami receptor components T1R1 or T1R3. We also demonstrated that with lactisole pretreatment, denatonium increased cytosolic Ca^2+^ signaling and downstream NO production. Coupled with our previous findings that amino acids reduce denatonium-induced apoptosis [25], together these results suggest that the combination of lactisole and denatonium may provide a therapeutic approach to activating NO production without unwanted cell death.

## 4. Discussion

Lactisole, an inhibitor of both umami and sweet signaling pathways, has previously been reported to bind to T1R3 and inhibit T1R3-mediated Ca^2+^ elevations in response to sugars or amino acids [11,27,29]. Umami and sweet tastes are mediated by heteromeric T1R1/T1R3 and T1R2/T1R3, respectively. These two taste modalities are perceived as “pleasant” because they signal the presence of beneficial amino acids or sugars, respectively [13]. However, T1R receptors serve as nutrient sensors all over the body, including in adipocytes [4], pancreatic β cells [5,6,7,8,9,29], and airway epithelial cells [19]. These extraoral T1Rs detect glucose or other sugars as well as amino acids to regulate physiological responses. Consequentially, their pharmacological manipulation has been proposed as a therapeutic modality for diseases such as diabetes [5,6,7,8,9] and chronic rhinosinusitis [19]. This study reveals novel insights into off-target effects of the most commonly used T1R inhibitor, lactisole. These effects must be taken into account if/when lactisole is used to modify T1R signaling. 

We have previously shown that in our basal airway epithelial model, at a physiological pH, umami agonists do not activate Ca^2+^, but instead signal a cAMP elevation. To our surprise, lactisole itself increased cAMP. Through our knockdown models, we showed that lactisole activates cAMP in a dose dependent manner through a mechanism independent of T1R1 or T1R3. Though the mechanism of this off-target activity will have to be explored in future work, here we show that through regulating ER Ca^2+^ levels and cytosolic Ca^2+^ release pathways, lactisole may provide a useful function to therapies utilizing bitter compounds to signal NO production. Our results also suggest that using lactisole in experiments should be approached with caution, as care must be taken to elucidate if the effects observed are truly due to T1R3 inhibition or due to off-target cAMP elevation. To our knowledge, off-target effects of lactisole have not been taken into account in previous papers using this compound as a T1R3 inhibitor. We believe this is the first demonstration of T1R-independent effects of lactisole, which have largely remained unstudied.

Here, we showed that lactisole treatment increased ER Ca^2+^ content. Additionally, increased Ca^2+^ elevations were observed using other T2R ligands as well as histamine. Lactisole increased Ca^2+^ signaling from many bitter compounds including diphenhydramine, flufenamic acid, thujone, quinine, and PTC. It is important to note that Beas-2Bs express both PTC-sensitive and non-sensitive variations of T2R38 at unknown quantities. Lactisole may “enhance function” of a cell line that can partially detect PTC. Future experiments will be necessary to determine the specific signaling pathways causing this increase in ER Ca^2+^. From our observations utilizing ER Tracker, we hypothesize that lactisole may increase ER size. However, it is important to note that in the 1 h incubation with lactisole, the sulfonylurea receptor, which is targeted by ER Tracker, may be downregulated. While our work offers a brief insight into the possible effect of lactisole on ER structure, future, more detailed work investigating ER size, stress, and total ion content would be needed to fully understand the impact that lactisole is has on the ER. Overall, through modulation of ER Ca^2+^, lactisole increased the elevations of Ca^2+^ signaling via every stimulus we tested.

ER Ca^2+^ content is established by balancing the influx and efflux of Ca^2+^ to and from the ER. This is accomplished through regulating SERCA pump activity to increase ER Ca^2+^ uptake, channels or receptors that leak/release Ca^2+^ from the ER, and chaperones that bind Ca^2+^ [44,45]. EPAC has been reported to increase Ca^2+^ mobilization through the activity of SERCA [37] and type 2 ryanodine receptor [38] to trigger a calcium-induced calcium release pathway [39,40]. In this pathway, the release of Ca^2+^ drives further Ca^2+^ release. However, as shown here, airway epithelial cells do not express type 2 ryanodine receptor transcript and do not elevate intracellular Ca^2+^ in response to 10 mM caffeine, a ryanodine receptor agonist [46]. Therefore, these receptors are most likely not contributing to any Ca^2+^ release pathways. Here, we have shown that EPAC agonist 2′-O-Me-cAMP increased ER Ca^2+^ efflux, causing a greater release of Ca^2+^ when cells were treated with 15 mM denatonium. Additionally, we have shown that 2′-O-Me-cAMP may also have a role in signaling extracellular Ca^2+^ uptake into the cell. Thus, in basal airway epithelial cells, we hypothesize that lactisole’s cAMP generation may be, at least in part, contributing to EPAC activation as both lactisole and 2′-O-Me-cAMP increase ER Ca^2+^ efflux.

Lactisole has an additional effect of increasing ER Ca^2+^ stores that 2′-O-Me-cAMP does not have, which suggests that SERCA activity may also be increased to offset the increase in ER Ca^2+^ efflux. Lactisole also increased the number of cells responsive to Ca^2+^ signaling stimuli which we did not observe with 2′-O-Me-cAMP. We have previously shown that treatment with 100 µM isoproterenol or 20 µM forskolin did not increase ER Ca^2+^ content in Beas-2Bs but did increase the number of cells responsive to stimuli [25]. Therefore, the increase in ER Ca^2+^ content may be due to a pathway that is independent of cAMP, PKA, or EPAC, while the increase in the number of responsive cells may either be due to a low level of PKA activation that is undetected by our biosensor or an undefined cAMP-signaled pathway.

Interestingly, lactisole treatment caused an increased cytosolic Ca^2+^ release in response to T2R agonist denatonium. We have previously shown that denatonium signals minimally through cytosolic Ca^2+^ [24]. Here, we observed that lactisole pretreatment greatly elevates denatonium’s intracellular Ca^2+^ signaling pathways. This Ca^2+^ elevation was specific to the cytosolic compartment as neither nuclear Ca^2+^ nor mitochondrial Ca^2+^ increased the above non-lactisole-treated cells. Hence, it is unlikely that this Ca^2+^ is a “bleed-through” from nuclear or mitochondrial signaling pathways but may utilize a cytosolic-specific pathway for Ca^2+^ efflux from the ER. It is also possible that downstream signaling components activated by lactisole may be causing post-translational modifications to cytosolic-specific IP3Rs. It may also be likely that there are simply more ER-to-cytosol Ca^2+^ efflux proteins and thus there is an overall greater flow of Ca^2+^ to the cytosol than any other compartment. In either case, our findings with lactisole suggest that because there is a greater reservoir of ER Ca^2+^, any treatment that stimulates ER Ca^2+^ release causes a greater burst of Ca^2+^ from the ER.

This pathway could be an important therapeutic target as this cytosolic Ca^2+^ elevation increased NO production in our model. Thus, bitter compounds such as denatonium may provide for important therapies through both utilizing lactisole to increase NO production and amino acids to reduce apoptosis. Given that lactisole increased ER Ca^2+^ over a prolonged period, these types of therapeutics may excel in treatments such as nasal lavage, where epithelial cells would be briefly exposed to a high concentration of lactisole and denatonium, while the remaining post-rinse residue may drive NO production.

In addition to the potential therapeutic aspects presented here, it is important to emphasize that nutrient sensing by T1Rs has been proposed to have multiple roles in human disease. One example is glucose sensing by T1R2/3 or T1R3 homodimers in pancreatic β cells, which may be important in diabetes [5,6,7,8,9,29]. Another example is the T1R2/3-mediated detection of glucose by solitary chemosensory cells in the nose [19]. The results above have important implications for attempts to manipulate sugar or amino acid detection by T1Rs with lactisole or derivatives. Off-target effects of lactisole on cAMP or subsequent changes in ER Ca^2+^ storing content and release may all have effects on other nutrient-sensing pathways that may confound the results obtained. This may require the use of T1R knockdown and/or knockout models rather than relying solely on pharmacological inhibition.

## 5. Conclusions

T1Rs are important nutrient sensors in many different cell types throughout the body. They detect amino acids (T1R1/3) and/or sugars (T1R2/3 or T1R3 homodimers) to regulate cell physiology that affects diverse processes such as taste and insulin secretion. Many studies have described the function of lactisole as an inhibitor of T1R3 and have used lactisole as a tool to pinpoint role of T1Rs. Here we showed a novel, uncharacterized function for lactisole that seemingly impacts all downstream Ca^2+^ signaling pathways through the regulation of ER Ca^2+^ storage. This off-target pathway is independent of T1Rs. Thus, lactisole may have the potential to augment the impact of any therapeutic that signals via downstream Ca^2+^ signaling pathways and may alter nutrient-sensing pathways even beyond the inhibition of T1Rs. 

We also showed that these off-target effects of lactisole might be therapeutically leveraged in airway epithelial cells. Intracellular Ca^2+^ modulates NO production [43]. NO is an important antimicrobial agent, and targeted therapies to initiate the host innate immune response in diseased patients have become highly sought after. We have found that in basal airway epithelial cells, which hyper-proliferate in disease states, bitter compounds signal both NO production and apoptosis [24]. We previously showed that with the addition of amino acids, denatonium-signaled apoptosis is nearly eliminated [25]. Moreover, the addition of lactisole amplified cytosolic Ca^2+^-signaled NO production in HBSS containing amino acids. Therefore, both amino acids and lactisole may be useful to employ in conjunction with bitter compounds to drive NO production to help combat infections.

## Figures and Tables

**Figure 1 nutrients-15-00517-f001:**
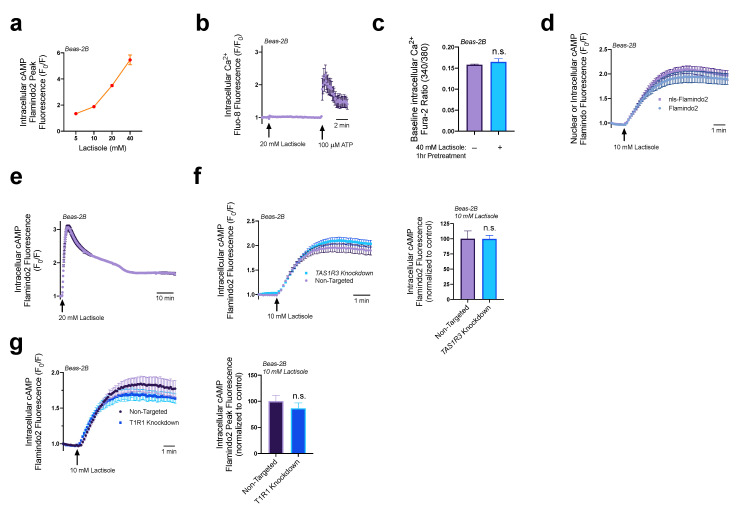
Lactisole stimulates intracellular cAMP elevations. (**a**) Beas-2Bs expressing cAMP biosensor Flamnido2 show a dose-dependent increase in cAMP in response to 5–40 mM of lactisole. (**b**) The 20 mM lactisole does not increase intracellular Ca^2+^ in Beas-2Bs (**c**) Beas-2Bs treated with 40 mM lactisole for 1 h show no difference in baseline intracellular Ca^2+^ as measured by Fura-2 fluorescence. (**d**) The 10 mM lactisole increases both intracellular and nuclear cAMP in Beas-2Bs. (**e**) cAMP elevations lasting for an hour from lactisole treatment. (**f**) There were no significant differences in lactisole-induced cAMP elevations in either T1R3 knockdown or non-targeted control cultures. (**g**) Knocking down T1R1 also had no significant impact on lactisole-mediated cAMP elevations. Traces are representative results from ≥3 experiments. Bar graphs show mean ± SEM from ≥3 experiments. Significance determined by *t*-test, “n.s.” represents no significance.

**Figure 2 nutrients-15-00517-f002:**
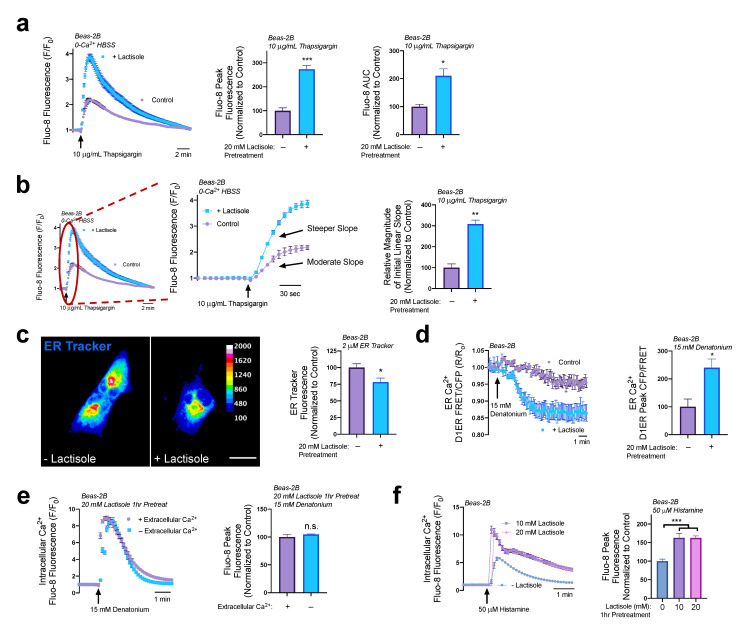
Lactisole increases ER Ca^2+^ content and size. (**a**) The 20 mM lactisole 1 h pretreatment increases peak Ca^2+^ release by 275% and ER Ca^2+^ content by 200%. (**b**) Analyzing the initial linear slope from trace in experiment (**a**), as highlighted by red circle and enlarged in graph on the right, revealed that lactisole treatment (20 mM, 1 h) caused a 300% steeper slope relative to untreated cells. (**c**) Beas-2B cells treated with 20 mM lactisole for 1 h revealed a decrease in ER tracker fluorescence. (**d**) Beas-2Bs expressing ER Ca^2+^ sensor D1ER show that denatonium treatment reduced ER Ca^2+^ stores 250% with 20 mM lactisole pretreatment (1 h). (**e**) Lactisole increases denatonium-induced Ca^2+^ release regardless of extracellular Ca^2+^ (**f**) Both 10 and 20 mM lactisole pretreatment (1 h) increases histamine-induced Ca^2+^ release by 50%. Traces are representative of ≥3 experiments. Bar graphs containing two comparisons were analyzed via *t*-test; bar graphs containing >2 comparisons were analyzed via ANOVA using Bonferroni’s post-test for multiple comparisons * *p* < 0.05, ** *p* < 0.01, *** *p* < 0.001.

**Figure 3 nutrients-15-00517-f003:**
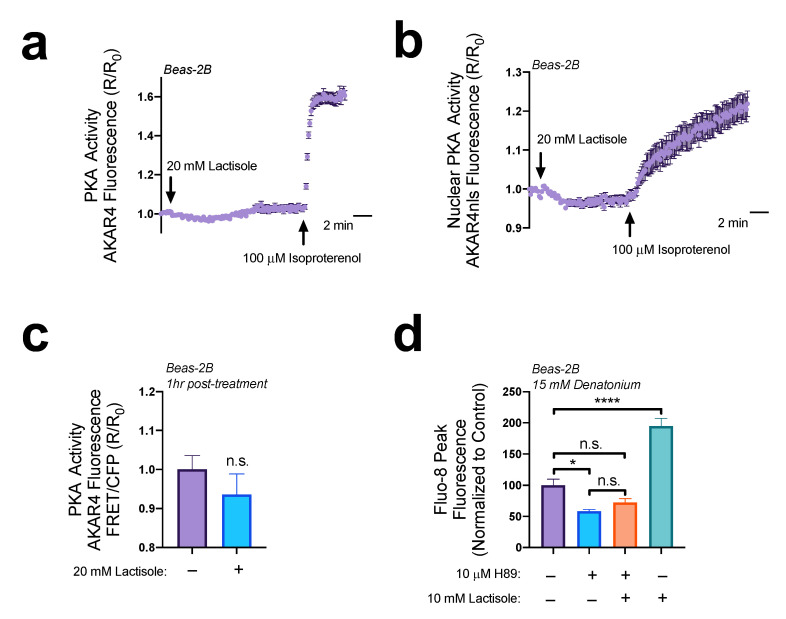
Lactisole cAMP elevations may not activate PKA. (**a**) The 20 mM lactisole did not alter PKA activity in Beas-2Bs expressing PKA biosensor AKAR4. (**b**) The 20 mM lactisole did not alter nuclear PKA activity in Beas-2Bs expressing nuclear PKA biosensor AKAR4nls. (**c**) lactisole (20 mM) did not alter baseline PKA activity even after a 1 h pre-treatment. (**d**) that PKA inhibitor H89 (10 µM, 1 h pretreatment) greatly impairs denatonium-induced Ca^2+^ release even with 20 mM lactisole pre-treatment (1 h). Bar graphs containing 2 comparisons were analyzed via *t*-test; bar graphs containing >2 comparisons were analyzed via ANOVA using Bonferroni’s post-test for multiple comparisons * *p* < 0.05, **** *p* < 0.0001, “n.s.” represents no significance.

**Figure 4 nutrients-15-00517-f004:**
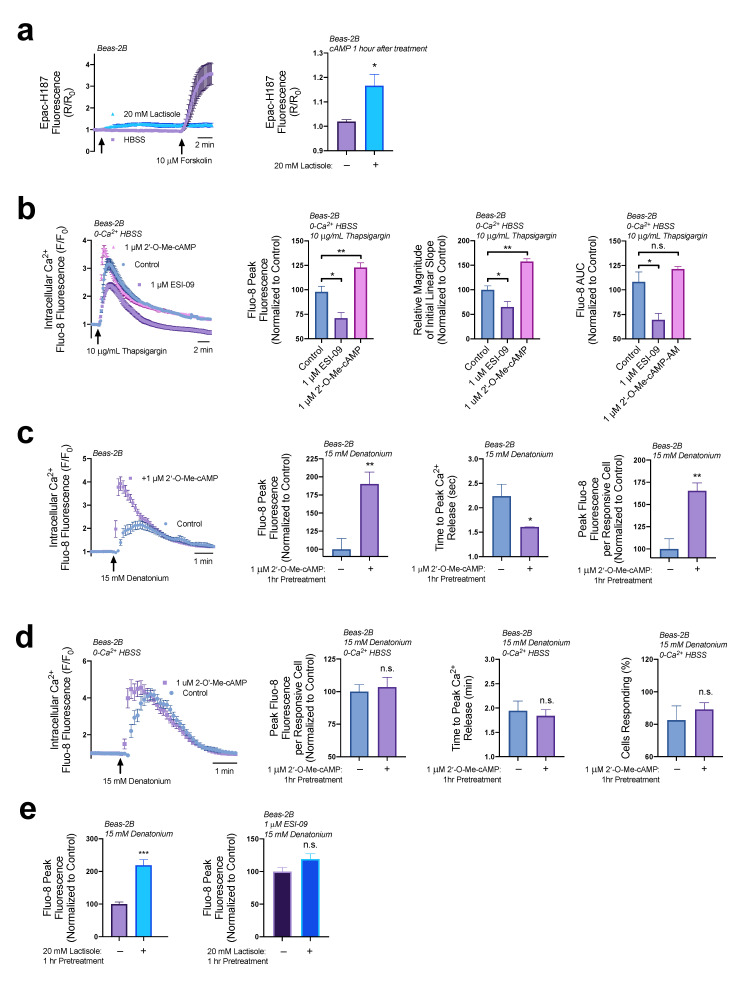
EPAC regulates ER Ca^2+^ content and efflux. (**a**) The 20 mM lactisole treatment caused a sustained increase in cAMP as detected by Epac-S-H74 biosensor. (**b**) EPAC agonist 2′-O-Me-cAMP (1 µM, 1 h pretreatment) increased peak ER Ca^2+^ release by 25%, increased the magnitude of the initial linear slope by 50% but had no effect on total ER Ca^2+^ content while antagonist ESI-09 (1 µM, 1 h pretreatment) decreased peak ER Ca^2+^ release by 25%, decreased the magnitude of initial linear slope by 35%, and reduced total ER Ca^2+^ content by 30%. (**c**) EPAC agonist 2′-O-Me-cAMP (1 µM, 1 h pretreatment) increased denatonium-induced Ca^2+^ elevations by 1.8-fold, reduced the time it took to reach maximal Ca^2+^ elevations by 30 s, and increased the Ca^2+^ released per cell by 67%. In a similar experiment as (**d**) Beas-2Bs preincubated with 2′-O-Me-cAMP (1 µM, 1 h) were stimulated with 15 mM denatonium in 0-Ca^2+^ HBSS, revealing no significant changes in peak Ca^2+^ elevations per cell or time to peak Ca^2+^ elevation. (**e**) The 20 mM lactisole (1 h pretreatment) increases denatonium-induced Ca^2+^ release by ~200% but is blocked by EPAC inhibitor ESI-09 (1 µM for 1 h pretreatment). Traces are representative of ≥3 experiments. Bar graphs containing 2 comparisons were analyzed via *t*-test; bar graphs containing >2 comparisons were analyzed via ANOVA using Dunnett’s post-test for multiple comparisons * *p* < 0.05, ** *p* < 0.01, *** *p* < 0.001, “n.s.” represents no significance.

**Figure 5 nutrients-15-00517-f005:**
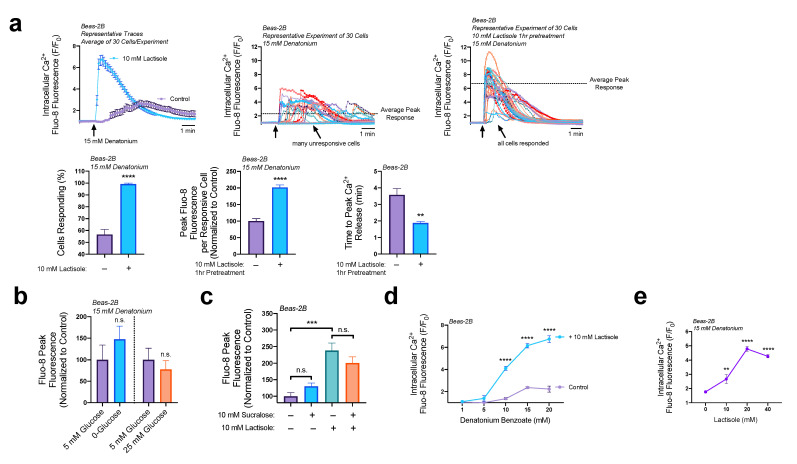
Lactisole increases denatonium induced Ca^2+^ release. (**a**) Lactisole (10 mM, 1 h pretreatment) greatly increased denatonium induced Ca^2+^ release, doubled the percentage of cells that responded to treatment with denatonium, doubled the peak Ca^2+^ release per cell, and reduced the time to peak Ca^2+^ release by 1.5 min. (**b**) Denatonium-induced Ca^2+^ release was unaffected by glucose concentrations. (**c**) Denatonium-induced Ca^2+^ release is unaffected by sweet taste receptor agonist sucralose. (**d**) Lactisole increases Ca^2+^ release from various concentrations of denatonium. (**e**) The 20 mM of lactisole has an optimal effect on denatonium induced Ca^2+^ signaling pathways. Bar graphs of only 2 comparisons were analyzed by Student’s *t*-test, bar graphs containing >2 data points were analyzed via ANOVA using (**c**) Tukey’s post-test for multiple comparisons, (**d**) Sidak’s post-test for paired comparisons, or (**e**) Dunnett’s post-test for comparison with control lacking lactisole: ** *p* < 0.01, *** *p* < 0.001, **** *p* < 0.0001, “n.s.” represents no significance.

**Figure 6 nutrients-15-00517-f006:**
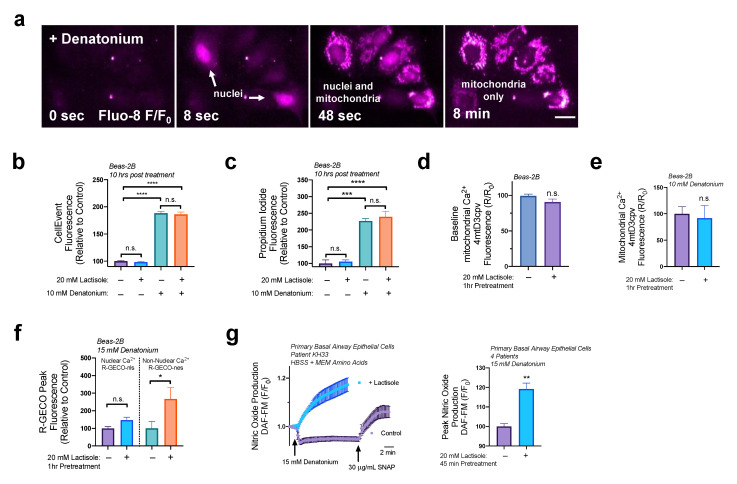
Lactisole increases denatonium-induced cytosolic Ca^2+^ elevations and NO production. (**a**) Fluorescence images showing time course of denatonium-induced Ca^2+^ elevations in starting nuclei then persisting in the mitochondria in airway epithelial cells; original image summarizing our findings from [24]. (**b**) The 20 mM lactisole pretreatment (1 h) does not alter caspase 3/7 activity initiated by denatonium signaling pathways. (**c**) The 20 mM lactisole pretreatment (1 h) does not alter cell death initiated by denatonium signaling pathways as detected by propidium iodide staining. (**d**) No changes in baseline mitochondrial Ca^2+^ levels with 20 mM lactisole pretreatment (1 h) were observed in Beas-2B cells. (**e**) There were no significant changes in denatonium-induced mitochondrial Ca^2+^ elevations with 20 mM lactisole pretreatment (1 h) in Beas-2B cells. (**f**) The 20 mM lactisole pretreatment (1 h) increased denatonium-induced non-nuclear Ca^2+^ elevations and had no effect nuclear Ca^2+^ elevations. (**g**) Primary basal airway epithelial cells in the presence of 1x MEM AA and with 20 mM lactisole pretreatment (1 h) initiate NO production in response to 15 mM denatonium. Traces are representative of ≥3 experiments. Bar graphs containing two comparisons were analyzed via *t*-test; bar graphs containing >2 comparisons were analyzed via ANOVA using Bonferroni’s post-test for multiple comparisons * *p* < 0.05, ** *p* < 0.01, *** *p* < 0.001 **** *p* < 0.0001, “n.s.” represents no significance.

## Data Availability

Data and/or reagents are available upon request to R.J.L., rjl@pennmedicine.upenn.edu.

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
