# Peer review of "Utilizing the Off-Target Effects of T1R3 Antagonist Lactisole to Enhance Nitric Oxide Production in Basal Airway Epithelial Cells"

_nutrients, 2023, doi:10.3390/nu15030517_

Round 1

Reviewer 1 Report

1. How does lactisole pretreatment 532 highly elevates denatonium’s intracellular Ca2+ signaling pathways? Elaborate the mechansism involve. 

 2. How pathway could be an important therapeutic target  (LINE: 540)? Highlight the the therapeutical approaches that could be achieved. 

3. Maintain uniformity in entire manuscript eg. Nitric oxide (NO) why using nitric oxide every must use mentioned abbreviation. LINE: 19, 47, 81 ... Check entire manuscript.

4. Kindly check the flow of contents it seems some places it is lacking. 

5. Make entire manuscript references uniform eg. 23 

Reviewer 2 Report

This paper is serial report of the authors for ectopic function of taste receptors and their intracellular signaling. In the past, the authors already reported the bitter receptors which detect bacterial derived metabolites and induce defensive responses in airway, the antagonistic effects of sweet/umami taste receptors for bitter receptors. Meanwhile, most of the pharmacologic approach to test sweet taste receptors function, T1R3 (the coreceptor subunit for sweet/umami receptor) inhibitor, lactisole has been used popularly. This study demonstrated that lactisole not only function as an inhibitor for T1R3, but also it harbors off-target effects for ER physiology via unknown molecular mechanism, implying that previous pharmacologic intervention using lactisole should be reconsidered. This finding is worthy for taste receptor research field. Moreover, based on the variety fluorescent reporter panels, this study is well-designed and display faithful results to draw a conclusion. Thus, I suggest that this article is enough to be published at Nutrient, but following issues should be addressed and corrected.

1. Most impressive effects of lactisole seems to be the expansion of ER size. The authors fully address the other effect of lactisole for facilitating Ca2+ efflux, and its mechanism in detail, but could not explain the ER size expansion. Even though ER-Tracker Green was used to observe the morphologic changes of ER, they should address whether the entire physical size of ER pool was increased or not, because the agent relies on its affinity against sulfonylurea receptor. To rule out whether decrease of fluorescent signals (in their intensity and size) just represent the sulfonylurea receptors downregulation, at least western blot should be conducted. Moreover, I suggest ultrastructural observation to see whether the lactisole treatment reorganize the ER structure.

2. Have you ever checked that PKA inhibitor H89 induced in changes of AKAR4 fluorescent? I think that omission of appropriate positive control misleads your interpretation. As you used H89 as selective inhibitor for PKA, it might serve as positive controls for the AKAR4 imaging experiment. If you could not detect any changes upon H89 treatment, you can conclude that at least in Beas-2B cells AKAR4 is not working, and exclude the experimental sets from the manuscript.

3. Previously, a lot of papers used lactisole to inhibit the sweet taste receptors pharmacologically. As the off-target effect of lactisole has been neglected so far, the potential impact of your novel finding should be address in the conclusion section.

4. There are minor typos.

- In line 212, “sulphonylurea” should be corrected to “sulfonylurea”

- In line 232, “this additional denatonium-induced Ca2+ release”.

Until the Figure 2C, and just before this phrase, you did not mention denatonium experiment elsewhere. I think that the order of sentences in this paragraph was shuffled during the editorial step. Please correct this.

-In this manuscript, the authors mix to use “1 hour” and “1 hr”. Please, unify the notation.

-In line 384, : “addition of denatonium doubled the number of cells responding to the denatonium treatment”

I think that the first “denatonium” is typo for “lactisole”. Please check this.
